# Structure-Property Relationships of 2D Ga/In Chalcogenides

**DOI:** 10.3390/nano10112188

**Published:** 2020-11-02

**Authors:** Pingping Jiang, Pascal Boulet, Marie-Christine Record

**Affiliations:** 1Aix-Marseille University, CNRS, MADIREL, 13013 Marseille, France; pingping.jiang@etu.univ-amu.fr; 2Aix-Marseille University, Université de Toulon, CNRS, IM2NP, 13013 Marseille, France; m-c.record@univ-amu.fr

**Keywords:** two-dimensional materials, DFT calculations, topological property, vdW homo- and heterostructure, structure-property relationship

## Abstract

Two-dimensional MX (M = Ga, In; X = S, Se, Te) homo- and heterostructures are of interest in electronics and optoelectronics. Structural, electronic and optical properties of bulk and layered MX and GaX/InX heterostructures have been investigated comprehensively using density functional theory (DFT) calculations. Based on the quantum theory of atoms in molecules, topological analyses of bond degree (BD), bond length (BL) and bond angle (BA) have been detailed for interpreting interatomic interactions, hence the structure–property relationship. The X–X BD correlates linearly with the ratio of local potential and kinetic energy, and decreases as X goes from S to Te. For van der Waals (vdW) homo- and heterostructures of GaX and InX, a cubic relationship between microscopic interatomic interaction and macroscopic electromagnetic behavior has been established firstly relating to weighted absolute BD summation and static dielectric constant. A decisive role of vdW interaction in layer-dependent properties has been identified. The GaX/InX heterostructures have bandgaps in the range 0.23–1.49 eV, absorption coefficients over 10^−5^ cm^−1^ and maximum conversion efficiency over 27%. Under strain, discordant BD evolutions are responsible for the exclusively distributed electrons and holes in sublayers of GaX/InX. Meanwhile, the interlayer BA adjustment with lattice mismatch explains the constraint-free lattice of the vdW heterostructure.

## 1. Introduction

Two-dimensional (2D) materials with van der Waals (vdW) interlayer interactions are predicted to expand the emerging thin-film family in electronics and optoelectronics fields due to their unique planar and layer-dependent properties [1,2,3,4,5]. The rising physical and chemical mechanisms of 2D vdW materials spark novel structure–property explorations at the atomic scale [6,7,8,9]. Except for transition metal dichalcogenides [10,11,12], such as Mo and W dichalcogenides [12,13,14,15], group III-VI materials have gradually attracted attention in photovoltaic (PV) fields and hence have been studied experimentally and theoretically [16,17,18,19]. Demirci et al. [19] found that III–VI monolayers are thermally stable with adequate stiffnesses and rigidities and of wide-spreading bandgaps. Coupling with high carrier mobility and optical susceptibility [18,20,21], their bright perspective in tunable and flexible device applications has been suggested. 

Chen et al. [17] found that the intrinsic mobility in III–VI monolayer is limited by the phonon scattering mode. In particular, with interfacial suppression of carrier scattering, the carrier mobility of multilayer InSe could reach 1055 cm^2^ V^−1^ s^−1^ according to Feng et al. [22]. Jappor and Habeed [23] have confirmed GaS and GaSe monolayers as promising materials in solar cells due to their high refraction indexes. Layer-dependent electronic and dielectric properties of Ga chalcogenides and mechanically tunable bandgaps in monolayers make them ideal candidates for nanoelectronics and optics [24]. The InSe monolayers, which are synthesized on SiO_2_/Si and mica substrates by physical and chemical vapor deposition, respectively, are n-type with the on/off current ratio over 10^4^ and few orders of magnitude of optical second-order susceptibility higher than the MoS_2_ one [20,21]. The GaS and GaSe monolayers synthesized on mica substrate have high photo-responsivities, being prospective in photodetector and photocatalyst [25,26]. Meanwhile, dozens of heterostructures shall be created by binding different 2D materials together with vdW forces. Competitive 2D vdW heterostructures should have diverse electronic performances, favorable band alignments and high PV effects as results of mutual efforts of sublayers [27]. The GaSe/ and InSe/graphene ones are of interest in field-effect transistors and dual Schottky diode devices because of the broad-band transparency of graphene, appropriate bandgaps of GaSe and InSe monolayers and their suitable band lineups [28,29]. Also, Jappor et al. [30] found that the GaS/GaSe heterostructure has a direct bandgap with the value smaller than that of the constitutive monolayers. Regarding InS/GaSe, GaS/GaSe and GaSe/GaTe ones [16], their type-II interfaces yield separately distributed hydrogen and oxygen at opposite sublayers, in favor of conversion efficiency. 

To date, layer-dependent properties of homo- and heterostructures of Ga and In chalcogenides have been detailed, which are essential for sustainable energy exploitation that heralds nanoscale PV applications. However, the topological mechanisms of intralayer and interlayer bondings when vdW structures experience homo- and hetero-stackings, remain unknown. Also, the role of each bonding in participating in the structure–property relationship and the quantification of that participation stay unresolved. Based on the quantum theory of atoms in molecules (QTAIM) [31], topological discussion of electron density and Laplacian distributions and local energy densities contributes to interpreting microscopic interatomic interactions, hence establishing a connection to macroscopic electromagnetic behaviors. In this work, structural, electronic and optical properties of bulk, mono-, bi- and trilayered MX (M = Ga, In; X = S, Se, Te), and the nine GaX/InX heterostructures have been studied thoroughly. Influences of composition and dimensionality on interatomic interactions have been identified. A cubic relationship between weighted bond degree (defined in [32,33]) summation and static dielectric constant has been unraveled firstly. By modeling the constitutive GaX and InX bilayers with identical lattice constants and stacking orders to GaX/InX, the exact effect of lattice stackings on sublayers has been extracted. In consequence, reasons for GaX/InX having a constraint-free lattice and exclusively distributed electrons and holes in sublayers are given by examing the evolution of bonding features when subjected to lattice strain. Our research is of great bearing in understanding the structure–property relationship of III-VI materials, furthermore, shedding light on discovering promising 2D vdW materials.

## 2. Computational Details

First-principle calculations were carried out by the full potential linear augmented plane wave method (FP-LAPW), implemented in the program WIEN2k [34]. For bulk calculations, WC-GGA [35], mBJ [36] and optB88-vdW [37] functionals were adopted to describe the exchange–correlation energy. For bulk calculations, the first Brillouin zones were sampled with 1500 ***k***-points, which corresponds to the grid 20×20×3
***k***-mesh, using Monkhorst-Pack grids [38]. The convergence criteria of total energy and force were set to 10^−5^ Ry and 1 mRy/Bohr, respectively. After lattice optimization and relaxation, band structure calculations were performed, as plotted in Appendix A. Table 1 lists the calculated lattice constants and bandgaps of hexagonal P6_3_/*mmc* GaX and InX (X = S, Se, Te) as well as other theoretical and experimental data. The good agreement between our calculated lattice parameters and those from literature validates our choices for the technical parameters settings, which will be used for the subsequent calculations on layered and heterostructures despite the underestimated bandgaps with the WC-GGA and optB88-vdW functionals. The 4×4×1 supercells of mono-, bi- and trilayered GaX and InX were modeled, as shown in Figure 1a, with a 20 Å thickness of vacuum atop to avoid periodic interactions. The nine GaX/InX heterostructures were built by vertically stacking GaX and InX monolayers with the metal atoms from the top monolayer, which are placed atop the chalcogen atoms from the bottom monolayer. Their in-plane lattice constants were set to the average values of those in bulk GaX and InX, and the positional relaxations were calculated by both WC-GGA and optB88-vdW functionals. In supercell calculation, a 15×15×1
***k***-mesh was used. After structure relaxations, the topological properties of electron density were calculated by the program CRITIC2 [39], based on the partition of real space into basins. Each basin only has one nucleus and is delimited by zero-flux surfaces of electron density. The local total (*H*), kinetic (*G*) and potential (*V*) energy densities, as functionals of electron density, can be determined by integrating over the surface of each basin [31]. For more details about the QTAIM method applications refer to our previous works [40,41].

## 3. 2D Layer-Dependent Properties

### 3.1. Stability and Electronic Property

The atomic configurations of mono-, bi- and trilayered [MX]_n_ slabs with M = Ga, In and X = S, Se, Te when n equals to 1, 2 and 3 are shown in Figure 1a. The in-plane lattice constants have been taken from the bulk ones and the out-of-plane ones have been optimized by energy minimization, followed by positional parameter relaxations. To examine the stability of the modeled slab, the formation energy is employed, which is given by $E_{form}=E_{slab}-\underset{x=M,X}{\sum }N_{x}E_{x}^{bulk},$ where *E*_slab_ is the total energy of the slab and $N_{x},\;E_{x}^{bulk}\;$ are the atom number and bulk energy, respectively. The distances between metal (M) and chalcogen (X) atoms from the adjacent layers in [MX]_n=2_ and [MX]_n=3_, i.e. ⑤ d_M’–X_ and ⑨ d_M’’__–X’_, have been compared with those in bulks. The differences ($\Delta d_{n,bulk}$) as well as the above yield $E_{form}$ are gathered in Table 2. The layered [MX]*_n_* slabs are thermodynamically stable due to their negative $E_{form}$, and the interlayer distances are shortened because of their negative $\Delta d_{n,bulk}$ except for InSe bilayer and trilayer whose $\Delta d_{n,bulk}$ are small but positive. The possession of structural stability and reinforcement of binding strength between X–M–M–X units evidence the construction of single- and multilayered GaX and InX. With the thinning of layer thickness, deviations of physical and chemical properties from those in bulks will inevitably cause electronic and optical changes.

The band structures of bulk, mono-, bi- and trilayered [MX]_n_ have been plotted along the M-Σ-Γ-Ʌ-K-M direction with the WC-GGA functional, as shown in Figure 2. As can be seen, all mono-, bi- and trilayered structures have indirect bandgaps, though the bulk InX materials have direct ones. As the layer thickness *n* goes from 1 to 3, the electron transition energy decreases continuously and those of bulk structures are the lowest. This is opposite to the refractive indexes and absorption coefficients in Figure 1b,c, where GaS is given as an example. The dominant transition paths of GaS and GaTe are in Σ-M directions while those of GaSe, InS, InSe and InTe are in Σ-Γ directions. For all cases, the direct (Γ-Γ’, where “Γ’” indicates the position of conduction band maximum when it comes to the direct transition path) and indirect (Σ-Γ) transition energies are close, which is consistent with the results in Ref. [24]. As X goes from S to Te, the indirect bandgaps decrease and those of GaX are higher than those of InX for a certain X. In particular, unlike the zero bandgap of bulk InTe, mono-, bi- and trilayered InTe have nonzero ones, which proves the assumption that by varying dimensionality the MX slab could have a tunable bandgap. 

### 3.2. Topological Properties of Electron Density 

The distributions of electron density (*ρ*) and the Laplacian ($\nabla^{2}$*ρ*) of bulk, mono-, bi- and trilayered [GaS]*_n_* (as an example) have been determined, as plotted in Appendix A. The symbols “*b*”, “*c*” and “*r*” between atoms represent the bond, cage and ring critical points where electron flux is zero. Tracking the electron density gradient between nuclei *n*_1_ and *n*_2_ (in the form of “*n*_1_-b-*n*_2_”), the maximum *ρ* can be reached. The corresponding coordination “b” is defined as the bond critical point (BCP). At each BCP, the local total (*H*_BCP_), kinetic (*G*_BCP_) and potential (*V*_BCP_) energy densities can be obtained. The bond degree (BD = *H*_BCP_/*ρ*_BCP_ [32,33]) and the adimensional |*V*_BCP_|/*G*_BCP_ ratio are used to stand for the magnitude and the type of interatomic interaction, respectively, thus indicating the bonding nature of each considered system. Figure 3 shows the BDs vs. |*V*_BCP_|/*G*_BCP_ ratios and the bond lengths (BLs) of M–X (②, ④ and ⑧ in Figure 1a, M–M (①, ⑥ and ⑩ in Figure 1a and X–X (③ and ⑦ in Figure 1a, pairwise in layered [MX]*_n_*. From the local virial theorem [31] (*h*^2^/16π^2^*m*∇^2^*ρ*_BCP_ = 2*G*_BCP_ + *V*_BCP_, where *h* and *m* are Planck’s constant and the electron mass respectively, X–X pairwises lie in the pure closed-shell region where *H*_BCP_/*ρ*_BCP_ > 0 and |*V*_BCP_|/*G*_BCP_ < 1, presenting local charge-depletion interactions, i.e. forming vdW-like bonding. In contrast, M–M pairwises in [InSe(Te)]*_n_* and [GaS(Se,Te)]*_n_* and M–X pairwises in [MTe]_n_ lie in the pure shared-shell region where *H*_BCP_/*ρ*_BCP_ < 0 and |*V*_BCP_|/*G*_BCP_ > 2, presenting local charge-concentration interactions, i.e. forming covalent bonding. M-M pairwises in [InS]*_n_* and M–X pairwises in [MS(Se)]*_n_* lie in the *transit* closed-shell region where *H*_BCP_/*ρ*_BCP_ < 0 and 1 < |*V*_BCP_|/*G*_BCP_ < 2, forming the intermediate polar covalent interactions. For covalent interaction, its covalence degree correlates positively to the absolute BD. However, for non-covalent interaction, its softening degree correlates negatively to the absolute BD [32]. 

As shown in Figure 3a, the absolute BDs and |V_BCP_|/G_BCP_ ratios of X–X, M–M and M–X bonds in GaX material are higher than those in InX material, presenting the stronger covalent interaction but weaker vdW interaction in the former than in the latter. In particular, the BDs and |V_BCP_|/G_BCP_ ratios of X–X bonds in GaX and InX materials are linearly correlated with the coefficient of determination (R^2^) of 0.94. The BLs of X–X bonds in GaX materials are longer than those in InX materials, which is opposite to those of M–M and M–X bonds. As X goes from S to Te, the absolute BDs of X–X bonds decrease while those of M–X bonds first decrease, and then increase, though their BLs keep increasing. The exceptional cases are found for M–M bonds. The Ga–Ga and In–In bonds have incoherent BD and BL variations as X varies. Compared with X–X and M–X bonds, the BLs of M-M bonds are less significant. As the layer thickness goes from bulk to bilayer and to trilayer, the BDs of X–X bonds first increase then decrease to minima. Thus, the X–X interatomic interactions are weakened in bilayers while strengthened in trilayers compared with the counterparts in bulks. In the meantime, In–In bonds in layered [InX]*_n_* have stronger interatomic interactions than those in bulk InX, whereas Ga–Ga bonds in layered [GaX]*_n_* have weaker interatomic interaction than those in bulk GaX. This holds for M–Te bonds as well except M–S and M–Se ones whose BDs are overlapping as layer thickness varies. More evident BD variations are found for bulk and layered [MX]*_n_* with higher chalcogen atomic numbers. It is found that the BLs of all bonds remain unchanged with respect to the layer thickness, showing resistance to dimensionality.

From the above discussions, the dependence of interatomic interactions, electronic and optical properties to lattice structures, including composition and dimensionality, are confirmed. However, the relationship between microscopic interactions and macroscopic electromagnetic behaviors as well as the participation of vdW and covalent bonds in that relationship are unclear. In the QTAIM [31], the electron density distribution and interatomic interaction are correlated to single and group atomic contributions to molecular polarizability. Since all individual constituents contribute to the frequency-dependent polarizability [52,53,54], their summation is related to the dielectric constant, as defined in the Clausius-Mossotti relation [55,56]. Morita et al. [57] found that in a crystal, relative permittivity links microscopic chemical bonding to macroscopic electromagnetic response. Therefore, the absolute BD summation is related to molecular polarizability and presumably also to dielectric behavior. Thus, in the quest for a structure–property relationship, we have searched for a relation between bond degree summation and dielectric constant in the materials of interest. In the bulk and layered [MX]*_n_*, there are three types of bonds, i.e. M–X, M–M and X–X, except for monolayers, for which only M–X, and M–M exist. The roles of bonds in the structure–property relationship are quantified by weight coefficients *h*, *l*, and *m*, respectively. The absolute BD summation is given by *h*|BD|_M–X_ + *l*|BD|_M–M_ + *m*|BD|_X–X_. The static dielectric constant along the layer thickening direction *ε*_1_(0) is used for electromagnetic behavior. 

Through polynomial fitting *h*|BD|_M–X_ + *l*|BD|_M–M_ + *m*|BD|_X–X_ to *ε*_1_(0), the maximum coefficient of determination (R^2^) can be obtained by adjusting *h*, *l*, and *m* as the equation order goes from first to second and to third. At each equation order, by setting *m* as the number of X–X bonds in unit cells of bulk, bilayer and trilayer, the R^2^ with respect to the *h*/*m* and *l*/*m* ratios can be obtained, as plotted in Appendix A. In particular, the *m* is zero for monolayers since there is no X–X bond. It is found that a cubic equation $y=A+Bx+Cx^{2}+Dx^{3}$ could describe the relationship between the absolute BD summation and static dielectric constant accurately; the maximum R^2^ of bulk, bi- and trilayered [MX]*_n_* are 0.999, 0.999 and 0.974, respectively, when *h*/*m* is 0.2 and *l*/*m* is 0.1, as shown in Figure 4a–c. The corresponding fitting curves and fitting coefficients A, B, C and D are shown in Figure 4d. In addition, the maximum R^2^ of monolayered [MX]*_n_* is 0.232 when *h* = 0, and *l* = 1 (not shown here). Compared with M–X and M–M bonds, the X–X bond plays a key role in the structure–property relationship by sharing the most responsibility in dielectric function; that sharing of constants also applies in bulks, bilayers and trilayers, suggesting the universality of that cubic relationship in multilayered vdW GaX and InX materials. The *h*|BD|_M–X_ + *l*|BD|_M–M_ + *m*|BD| and ε_1_(0) are inversely correlated and both increase with layer thickness. For instance, the InX has a smaller BD summation than GaX, resulting in stronger refraction and absorption than the latter. Compared to layered [MX]*_n_*, the highest BD summation and *ε*_1_(0) are found for bulks. Therefore, through bonding engineering, namely, interatomic interaction adjustments, chances for achieving equivalent optical responses with bulks but with fewer materials are foreseen.

## 4. Heterostructure Properties

### 4.1. Stability and Electronic Property

The GaX/InX (X = S, Se, Te) heterostructures are modeled with the GaX and InX monolayers taken from their respective bilayers and denominated from N°1 to N°9. The in-plane lattice constants of heterostructures and bilayers are chosen as the average values of those of bulk GaX and InX in Table 1, resulting in lattice mismatches Δ*a*/*a*_0_ at interfaces, as shown in Figure 5a. After forces relaxation, the heterointerface binding energy (*E*_b_) can be obtained from $E_{b}=1/2A\left[E_{GaX/InX}-1/2\left(E_{GaX}+E_{InX}\right)\right]$, where $E_{GaX/InX}$, $E_{GaX}$ and $E_{InX}$ are total energies of heterostructure and of its corresponding top and bottom bilayers, respectively, and A is the interface area. As plotted in Figure 5b, except for N°3 (GaS/InTe), the $E_{b}$ of GaX/InX are in the range [−12.7;−97.1] meV/Å^2^ with WC-GGA functional and [−5.8;−55.4] meV/Å^2^ with optB88-vdW functional, presenting vdW binding forces. Combined with the Δ*a*/*a*_0_, it can be concluded that the stability of vdW binding heterostructure is independent of lattice mismatch since high stability unnecessarily corresponds to small lattice mismatch. The most negative *E*_b_ is found for N°3, which contrasts with its largest lattice mismatch of over 10%. 

Band structures and density of states (DOS) of the above nine heterostructures are calculated by the WC–GGA functional and are plotted in Figure 6. Bandgaps of GaX/InX heterostructures are within the range 0.23–1.49 eV, except for the zero eV of N°3. Most of the heterostructures have indirect bandgaps in the Σ-Γ direction, except for N°6 and N°7, which have small but direct bandgaps in the Γ-Γ’ direction. Heterostructures have smaller bandgaps than the constitutive top and bottom bilayers at equilibrium lattices (see in Figure 2). Compared with the zero bandgap of bulk InTe, the none-zero ones of N°6 GaSe/InTe and N°9 GaTe/InTe suggest that by reducing layer thickness and stacking with others, materials could yield tunable bandgaps, making them candidates for 2D thin-film PV applications. According to DOS, the valence band maximum (VBM) and conduction band minimum (CBM) of N°1, N°4 and N°7–8 are composed of Ga-s/p, X-p orbitals and In-s/p/d, X-p ones, respectively. This exclusive contribution suggests a built-in separation of electrons and holes in heterostructures, thus upon excitation electrons generated in the GaX monolayer will uniquely flow to the InX one. In contrast, the VBM and CBM of N°3 are composed of the Te-p orbital in the InX monolayer and Ga-s/p, S-p ones in GaX monolayer, respectively, forming the electron flow path from the former to the latter. Special cases are observed for N°2, N°5–6 and N°9 whose VBMs and CBMs are both composed of In-s/p/d and X-p orbitals in InX monolayers. Thus upon excitation, electrons will flow within them. 

The valence and conduction band offsets, i.e. VBOs and CBOs, are obtained by formulae $\Delta E_{v}^{GaX/InX^{\prime }}=\Delta E_{VBM,C}^{GaX}-\Delta E_{VBM,C^{\prime }}^{InX^{\prime }}+\Delta E_{C,C^{\prime }}$ and $\Delta E_{c}=\Delta E_{g}+\Delta E_{v}$, where $\Delta E_{VBM,C}^{GaX}$ and $\Delta E_{VBM,C^{\prime }}^{InX^{\prime }}$ are the energy differences between core levels (X/X’-1s) and VBMs in bulk GaX and InX, respectively, and $\Delta E_{C,C^{\prime }}$ is the binding energy difference between X-1s and X’-1s at the interfaces [41,58]. As plotted in Figure 7a, N°1, N°3–4 and N°7–8 heterostructures belong to type II (“cliff-like”) band offsets, where their VBOs and CBOs have opposite signs. Positive VBO coupled with negative CBO leads to the electrons jumping from GaX to InX, whereas negative VBO coupled with positive CBO brings the electrons jumping from InX to GaX. Fewer electron-hole recombinations are predictable at heterointerfaces. Indeed, large VBO and small CBO guarantee the separation of charge carriers in active layers by hindering their injections to the counterpart layers. For example, N°7 has a larger VBO but a smaller CBO than N°1, which will contribute to a superior ability in electron excitations and collections of the former than those of the latter. In particular, N°3 has the supreme VBO, −2.08 eV, and almost flat CBO, −0.04 eV, which in no doubt favors the PV ability under an external electric field. Numerous electrons generated in the InTe side could travel through the interface easily and enter the GaS side, while the left holes in the InTe side are incapable to surpass the interfacial boundary, thus reducing the electrons–holes recombinations and forming the electron flow path. The flat CBO could also explain its zero bandgap. The N°2, N°5–6 and N°9 heterostructures belong to type-I (“spike-like”) band offsets, sharing the same negative signs for VBOs and CBOs. Therefore, electrons generated in InX monolayers are highly constrained by the negative CBOs, while the holes keep flowing into the InX monolayer due to the negative VBOs. The consistency between DOS and band alignments could help to assess the electronic competence for individual heterostructure.

In order to assess the optical properties of the built GaX/InX, refractive indexes, absorption coefficients and loss tangents have been studied and compared, as shown in Appendix A. The GaX/InX has absorption coefficients over 10^5^ cm^−1^ and static refractive indexes over 2.0, which are in between those of the constitutive bilayers in Appendix A. The optical responses in the in-plane and out-of-plane directions are similar. Evaluation of the PV capacity of GaX/InX thin-film can be made by employing the spectroscopic limited maximum efficiency (SLME) method [59]. Based on the above out-of-plane absorption coefficient in the wavelength range 280–1200 nm, the short circuit current (*J*_SC_), open-circuit voltage (*V*_OC_) and thus conversion efficiency (*η*_max_) with respect to thin-film thickness can be obtained, as shown in Figure 7b,c. The AM 1.5G illumination is chosen as the input solar condition. With the growth of the film thickness, the fast increasing *J*_SC_ and slowly decreasing *V*_OC_, owing to charge carrier enrichment and recombination, shall yield a growing *η*_max_. The slopes of *J_SC_-V_OC_* curves of N°3, 6–9 are at a higher level than those of N°1–2, 4–5. Superior absorption ability is found for GaX/InX with a higher chalcogen atomic number, resulting in a faster converging *η*_max_. For instance, the *η*_max_ of N°3, 6–9 converge at 1.0 μm whereas those of N°1–2, 4–5 converge at 3.0 μm. Specifically, the converged *η*_max_ are 24.7% for N°1, 26.6% for N°2, 4–5, and 27.4% for N°3, 6–9, respectively. 

### 4.2. Topological Properties of Electron Density

Based on the distributions of electron density and the Laplacian, as shown in Appendix A, the bonding patterns in N°1–9 heterostructures, as well as those in their constitutive top and bottom bilayers, have been studied. For each bonding pairwise, its bond degree (BD) vs. |*V*_BCP_|/*G*_BCP_ ratio at BCP and the corresponding bond length (BL) have been plotted in Figure 8a–c, and Figure 8e–g, respectively. Figure 8d lists the lattice mismatch (*lm*) in GaX/InX and the resulting sign in the top [GaX]_2_ and bottom [InX]_2_, where “+” means tensile strain and “−” means compressive strain. The X–X bonds lie in the charge-depletion region with positive *H*_BCP_ and $\nabla^{2}$*ρ*_BCP_, whereas the M–M and M–X ones lie in the charge-accumulation region with negative *H*_BCP_ and either positive $\nabla^{2}$*ρ*_BCP_ or negative $\nabla^{2}$*ρ*_BCP_, which are similar to those in bulk and layered materials. The linear relationship between BDs and |*V*_BCP_|/*G*_BCP_ ratio of X–X bonds is maintained here as well. The BLs of X–X bonds in GaX/InX are within the range of 4.26–4.42 Å and irrelevant to the *lm,* which are close to the midpoint of those in the constitutive bilayers (dashed line in Figure 8e) except for N°3, i.e. GaS/InTe.

As the *lm* goes from “−” to “+”, BDs and BLs of S–S (N°1’’, 4’’, 7’’), Se-Se (N°2’’, 5’’, 8’’) and Te–Te (N°3’’, 6’’, 9’’) at the bottom [InX]_2_ and S–S (N°1’, 2’, 3’), Se–Se (N°4’, 5’, 6’) and Te–Te (N°7’, 8’, 9’) at the top [GaX]_2_ increase, as shown in Figure 8a,e. Thus, the X–X bonds are strengthened and shortened under compressive strain while weakened and lengthened under tensile strain since higher BD when *H*_BCP_/*ρ*_BCP_ > 0 corresponds to weaker interaction [33]. As X goes from S to Te, BDs and BLs of X–X bonds in N°1’’, 2’’, 3’’, in N°4’’, 5’’, 6’’ and in N°7’’, 8’’, 9’’ bottom [InX]_2_ decrease, while those of the corresponding X–X bonds in N°1’, 2’, 3’, in N°4’, 5’, 6’ and in N°7’, 8’, 9’ top [GaX]_2_ increase. This reverse bonding energy and bond length adjustments are reasons for X–X bonds in GaX/InX having BDs and BLs in between those in the individual top and bottom bilayers. By tuning the X–X interactions and geometry configurations, the lattice-constraint free virtues shall be preserved for vdW structures. Meanwhile, as *lm* goes from “−” to “+”, the absolute BDs and BLs of M–M and M–X bonds in [InX]_2_ and [GaX]_2_ decrease and increase, respectively, and their variations in the former are less significant than those in the latter. Thus, similar to X–X bonds, M–M and M–X bonds are strengthened and shortened under compressive strain while weakened and lengthened under tensile strain, and those in [InX]_2_ are less sensitive to strain than those in [GaX]_2_. For bonds in GaX/InX, the absolute BDs of In–In and In–X in their InX monolayers are respectively lower and higher than those in the corresponding [InX]_2_, whereas the absolute BDs of Ga–Ga and Ga–X in their GaX monolayers are close to those in the corresponding [GaX]_2_, as shown in Figure 8b–c. Therefore, after modeling the InX monolayer with the GaX one, In–In and In–X bonds are weakened and strengthened, respectively, while Ga–Ga and Ga–X bonds are barely changed, showing strong resistance to lattice stacking.

The ease of electron excitation can be scaled by the bond degree. A bond with high BD when *H*_BCP_/*ρ*_BCP_ > 0 means weak interatomic interaction, indicating excessive kinetic energy. Chances for electron excitations locating in between atoms are smaller than those with low BD. In the meantime, a bond with high absolute BD when *H*_BCP_/*ρ*_BCP_ < 0 means stronger interatomic interaction, which ends up with fewer electron excitations because of the excessive potential energy. Hence, extra energy is needed to produce equivalent electrons for both vdW and covalent bonds with high absolute BDs in regards to those with low absolute BDs. Compared with the S–S bond, the Te–Te bond is superior in electron excitation due to its lower BD, resulting in richer charge carriers in material. Given the strain-induced topological behaviors of the constitutive [InX]_2_ and [GaX]_2_, the electronic properties of GaX/InX are predictable. The N°1’’–6’’ and 9’’ X–X bonds in [InX]_2_ have lower BDs than N°1’–6’ and 9’ X-X bonds in [GaX]_2_. Robust electron excitations in InX monolayers of GaX/InX are foreseen. Electron transitions between atoms X are mainly contributed by X–p orbitals from the InX sides. After stacking the InX monolayer with the GaX one, the weakened In–In bonds coupling with the barely changed Ga–Ga bonds contribute to electron transitions between In-s/p orbitals. Therefore, the generated electrons in InX monolayers tend to flow within them. However, the greatly strengthened In–X bonds in N°1 and N°4 will significantly constrain electron transitions between In–p and X–p orbitals, leading to the large CBOs at interfaces. In consequence, electrons generated in GaX monolayers will exclusively flow to InX ones. The N°7’’–8’’ X–X bonds in [InX]_2_ have higher BDs than N°7’–8’ X–X bonds in [GaX]_2_. As a result, in GaX/InX the electrons will be generated in GaX monolayers and flow to InX ones as well. A special case has been found for the N°3. Its maximum BDs differences of X–X bonds in [InX]_2_ and [GaX]_2_ and of In–In bonds in [InX]_2_ and GaX/InX give rise to a strong tendency for generating electrons in the InX monolayer and flowing solely to the GaX one. 

At homo- and heterointerfaces, the bond angle (BA) of X–X in the form “*n*_1_–b–*n*_2_” (as mentioned before) are plotted in Figure 8h. As it shows, the values of X–X BA in [InX]_2_ and [GaX]_2_ are close to 180° and overlap with each other. However, those in GaX/InX are lower than 180° and are inversely related to lattice mismatches (see in Figure 5a). In contrast, the BAs of M–M and M–X bonds are barely changed before and after lattice stacking, as shown in Appendix A. The irrelevance of *E*_b_ to *lm* can be explained by the X–X BA adjustment. Under strain, the X–X bond encounters the *lm* by shifting BCP towards the atomic end with high electronegativity instead of yielding BL deflection. A larger *lm* would cause a greater BA deviation, in other words, a smaller value of BA, and vice versa. Consequently, the BLs could be maintained within a certain range. For example, N°2 GaX/InX has higher *lm* than the N°4 one, which corresponds to the smaller S–Se BA and similar S–Se BL of the former compared to those of the latter. This strain-correlated alignment between BA and *lm* accounts for the independence of interfacial stability to the lattice constraint of the vdW structure. 

The cubic relationship of microscopic interactions and macroscopic electromagnetic behavior, as established above, has been examined for GaX/InX heterostructures as well. The fitting of *h*|BD|_M–X_ + *l*|BD|_M–M_ +*m*|BD|_X–X_
*vs. ε*_1_(0) is proceeded by adjusting *h*/*m* and *l*/*m* ratios as equation order goes from first to second and to third. In particular, for GaX/InX unit cell, its BD summation is composed of *h*/2(|BD|_Ga–X_ + |BD|_In–X_), *l*/2(|BD|_Ga–Ga_ + |BD|_In–In_) and *m*|BD|_X–X_. As plotted in Figure 9a–c, the R^2^ increases with the equation order and the maximum R^2^ is 0.994 when *h*/*m* and *l*/*m* ratios are both 0.2. The applicability of that cubic relationship in vdW heterostructures is proved. As shown in Figure 9d, the absolute BD summation decreases as the chalcogen atomic number grows. This weakened interatomic interaction leads to stronger electron excitation, therefore to a higher optical response as a whole. Unlike the *h*/*m* ratio, the *l*/*m* ratio in GaX/InX is larger than it in bulk and multilayers, which means that under strain, M–M bonds in GaX/InX contribute more to structure-property relationships than those in structures without strain applied. Thus, the improvement of electromagnetic behaviors becomes possible by means of mechanical engineering.

## 5. Conclusions

Theoretical investigation on structural, electronic and optical properties of bulk, mono-, bi-, and trilayered MX (M= Ga, In; X = S, Se, Te) and GaX/InX heterostructures have been performed using DFT calculations. According to the QTAIM, topological properties of electron density and the Laplacian are used for interpreting interatomic interactions. Results show that X–X and M–M/–X bonds lie in charge-depletion and charge-accumulation regions, respectively. The BDs and |*V*|/*G* ratios of X–X bonds are linearly correlated. As chalcogen atomic number grows, X–X BD decreases, resulting in an increasing optical property. Layer-dependent electronic and optical properties have been examined. The relationship between microscopic interatomic interaction and macroscopic electromagnetic behavior has been quantified firstly by a cubic equation involving the absolute BD summation and static dielectric constant. The roles of X–X, M–M and M–X in that cubic relationship have been identified and that of X–X is decisive. The GaX/InX heterostructures have vdW binding forces with wide-spreading bandgaps, high absorption coefficients and maximum conversion efficiencies. The X–X, M–M and M–X are strengthened and shortened under compressive strain while weakened and lengthened under tensile strain. Discordant strain-induced topological properties in sublayers are responsible for the exclusive distributions of electrons and holes, yielding type-II band offsets. Under strain, the alignment between the interlayer bond angle and the lattice mismatch explains the constraint-free lattices of vdW heterostructures. 

## Figures and Tables

**Figure 1 nanomaterials-10-02188-f001:**
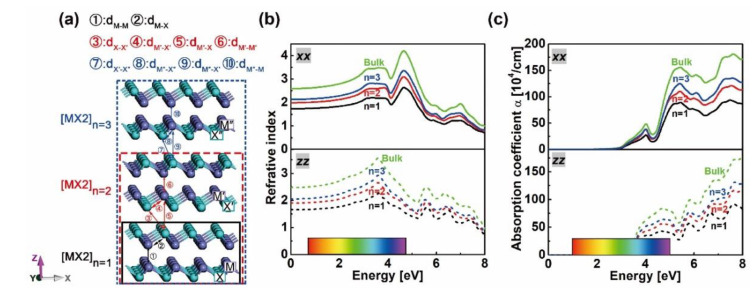
Schemes of [MX]n supercells with M = Ga, In and X = S, Se, Te slabs for n equals to 1, 2 and 3 respectively (**a**). Atoms from the first, second and third X–M–M–X unit in the relevant slabs are marked as [M,X], [M’,X’] and [M’’,X’’], respectively. Refractive index (**b**) and absorption coefficient (**c**) of bulk and layered [GaS]*_n_*.

**Figure 2 nanomaterials-10-02188-f002:**
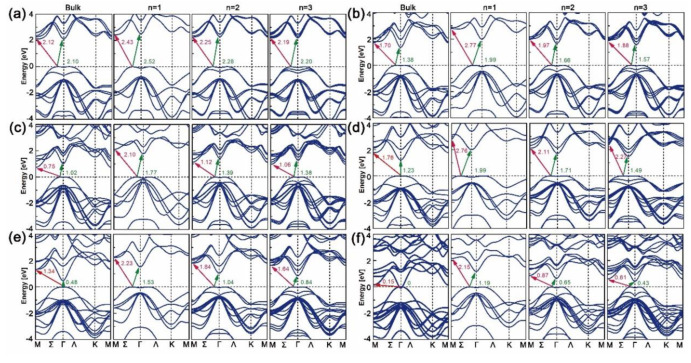
Band structures of bulk, mono-, bi- and trilayered (**a**) GaS, (**b**) GaSe, (**c**) GaTe, (**d**) InS, (**e**) InSe and (**f**) InTe calculated with WC-GGA functional.

**Figure 3 nanomaterials-10-02188-f003:**
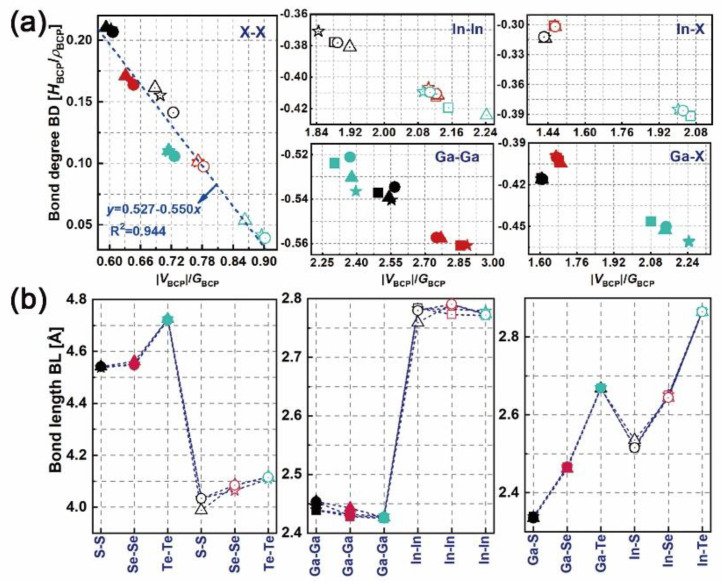
Bond degrees vs. the |*V*_BCP_|/*G*_BCP_ ratio: (**a**) and bond lengths, (**b**) of M–X (②, ④ and ⑧ in Figure 1a, M–M (①, ⑥ and ⑩ in Figure 1a and X–X (③ and ⑦ in Figure 1a bonds in bulk and layered [MX]*_n_*. Symbols used in plots, solid: GaX, hollow: InX; black, red and teal colors: X = S, Se and Te; squares, triangles, circles and stars: *n* = 1, 2, 3 and bulk.

**Figure 4 nanomaterials-10-02188-f004:**
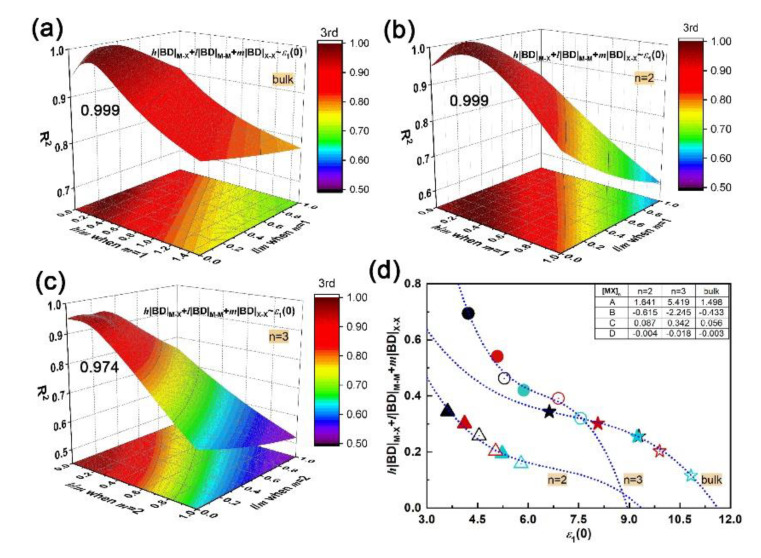
Fitting *h*|BD|_M–X_ + *l*|BD|_M–M_ + *m*|BD|_X–X_ vs*. ε*_1_(0) via equation $y=A+Bx+Cx^{2}+Dx^{3}$ by adjusting *h*/*m* and *l*/*m* ratios: (**a**) coefficient of determination (R^2^) of bulk; (**b**) bilayer; (**c**) trilayer; (**d**) fitting curves and coefficients A, B, C, D at the maximum R^2^. Symbols used in the (**d**) plot, solid: GaX, hollow: InX; black, red and teal colors: X = S, Se and Te, respectively; triangles, circles and stars: *n* = 2, 3 and bulk, respectively.

**Figure 5 nanomaterials-10-02188-f005:**
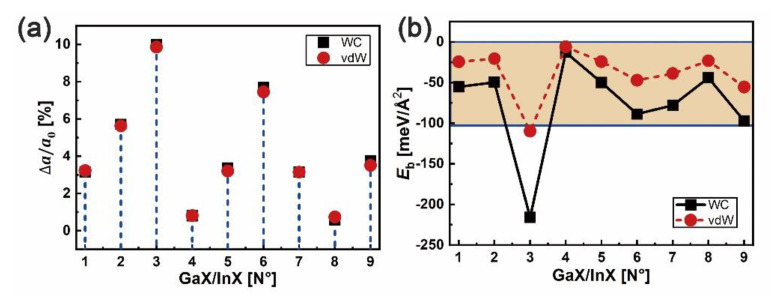
In-plane lattice mismatch Δ*a*/*a*_0_ in %: (**a**) and binding energy *E**_b_* in meV/Å^2^, and (**b**) at the GaX/InX (X = S, Se, Te) heterointerface.

**Figure 6 nanomaterials-10-02188-f006:**
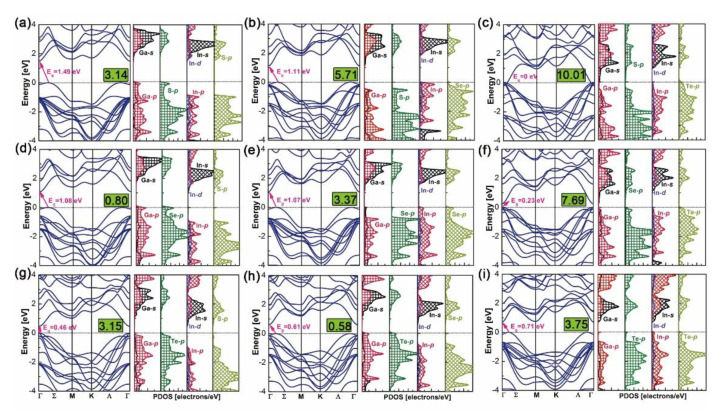
Band structures and density of states of N°1–9 GaX/InX (**a**–**i**). Numbers enclosed in green blocks are their in-plane lattice mismatches in %.

**Figure 7 nanomaterials-10-02188-f007:**
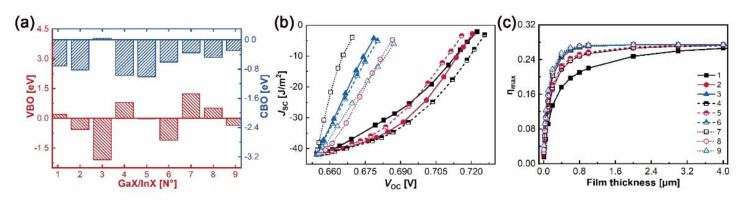
(**a**) Valence (red color) and conduction (blue color) band offsets; (**b**) calculated *J_SC_-V_OC_* curves and (**c**) conversion efficiency *η*_max_ under AM 1.5G illumination at different film thicknesses of GaX/InX.

**Figure 8 nanomaterials-10-02188-f008:**
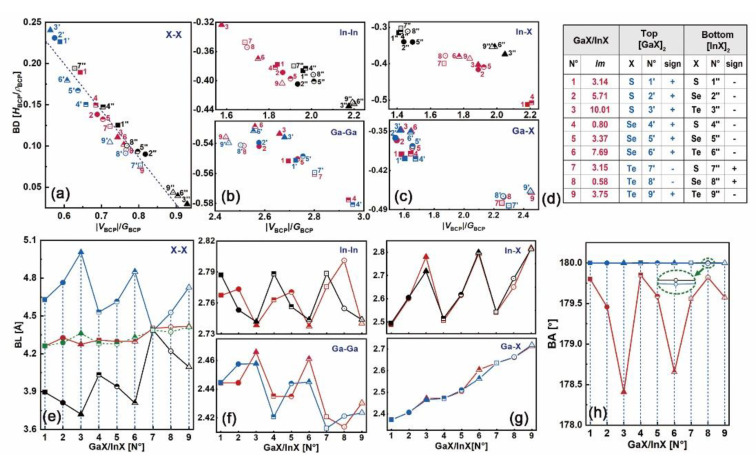
(**a**) BDs vs. |*V*_BCP_|/*G*_BCP_ at BCPs of X–X, (**b**) M–M and **(c)** M–X bonds. (**d**) Lattice mismatches (*lm*) in GaX/InX and the resulting signs in the respective top [GaX]_2_ and bottom [InX]_2_. (**e**) BLs of X–X, (**f**) M–M, and (**g**) M–X bonds. (**h**) Bond angle (BAs) of X–X bonds. Symbols in red, blue and black colors represent bonds in GaX/InX, top [GaX]_2_ and bottom [InX]_2_, respectively. solid square, circle, triangle: N°1–3, respectively; half-filled square, circle, triangle: N°4–6, respectively; hollow square, circle, triangle: N°7–9, respectively. Green dash line in **(e)** is the average X–X BL of [GaX]_2_ and [InX]_2_.

**Figure 9 nanomaterials-10-02188-f009:**
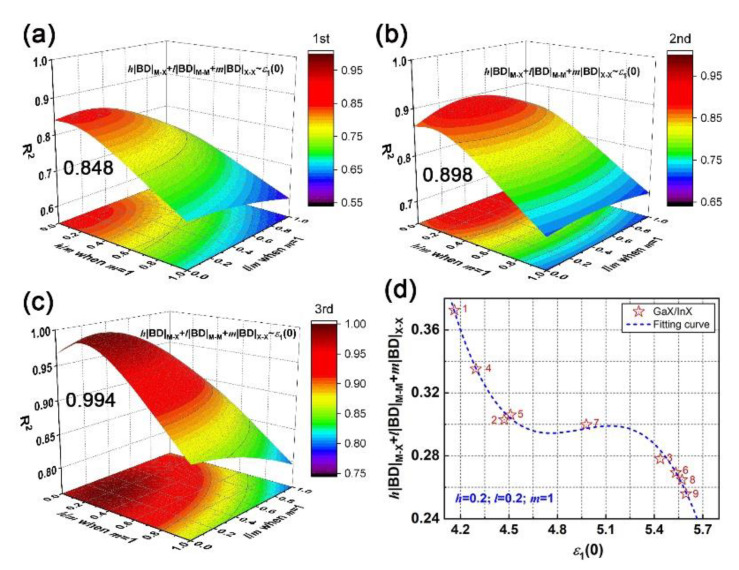
Fitting *h*|BD|_M–X_ + *l*|BD|_M–M_ + *m*|BD|_X–X_ vs. ε_1_(0) of N°1–9 GaX/InX by adjusting *h*/*m* and *l*/*m* ratios: the coefficient of determination (R^2^) as equation order goes from (**a**) first, (**b**), second, and (**c**) third. (**d**) The cubic fitting curve at the maximum R^2^ when *h*/*m* and *l*/*m* ratios are both 0.2.

**Table 1 nanomaterials-10-02188-t001:** Calculated lattice constants *a* and *c* and bandgaps of bulk MX (M = Ga, In; X = S, Se, Te) by using WC-GGA, optB88-vdW and mBJ functionals, as well as other theoretical “(c)” and experimental “(e)” data.

MX	*a* (Å)			*c* (Å)			*E*g (eV)			
	WC-GGA	optB88-vdW	Calc. & Exp.	WC-GGA	optB88-vdW	Calc. & Exp.	WC-GGA	optB88-vdW	mBJ	Calc. & Exp.
GaS	3.588	3.640	3.583 ^(c)^ [24] 3.587 ^(e)^ [42]	14.935	14.932	15.531 ^(c)^ [24] 15.492 ^(e)^ [42]	2.100	1.813	2.824	1.605 ^(c)^ [24] 2.530 ^(e)^ [43]
GaSe	3.760	3.821	3.751 ^(c)^ [24] 3.752 ^(e)^ [44]	15.488	15.467	15.948 ^(c)^ [24] 15.940 ^(e)^ [44]	1.384	1.095	2.476	0.995 ^(c)^ [24] 2.005 ^(e)^ [45]
GaTe	4.069	4.136	4.040 ^(c)^ [46] 4.060 ^(e)^ [46]	15.364	15.412	16.620 ^(c)^ [46] 16.960 ^(e)^ [46]	0.746	0.637	1.428	1.604 ^(c)^ [46] 1.650 ^(e)^ [47]
InS	3.820	3.884	-	14.238	14.253	-	1.226	0.960	2.313	-
InSe	4.022	4.075	3.940 ^(c)^ [48] 4.050 ^(e)^ [49]	15.113	15.085	16.850 ^(c)^ [48] 16.930 ^(e)^ [49]	0.484	0.367	1.449	1.2 ^(c)^ [50] 1.3 ^(e)^ [51]
InTe	4.385	4.437	-	15.043	15.058	-	0.000	0.138	0.786	-

**Table 2 nanomaterials-10-02188-t002:** Formation energy *E*_form_ of bulk and of [MX]*_n_* slabs when *n* equals 1, 2, and 3, and differences in the distance $\Delta d_{2,bulk}$ between M’–X in bilayer and bulk and $\Delta d_{3,bulk}$ between M’’–X’ in trilayer and bulk.

[MX]_n_	*E*_form_ (eV/motif)				N°⑤ (mÅ)	N°⑨ (mÅ)
	*n* = 1	*n* = 2	*n* = 3	bulk	∆*d*_2,bulk_	∆*d*_3,bulk_
GaS	−3.825	−7.655	−11.481	−7.611	−1.2	−20.7
GaSe	−1.684	−3.385	−5.105	−3.452	−2.0	−7.5
GaTe	−2.746	−0.691	−1.615	−1.823	−12.3	−29.4
InS	−3.637	−7.270	−10.913	−7.304	−0.8	−40.7
InSe	−2.235	−4.508	−6.788	−4.505	3.8	0.8
InTe	−3.370	−6.150	−9.776	−6.247	−10.4	−10.4

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
