# Peer review of "Structure-Property Relationships of 2D Ga/In Chalcogenides"

_nanomaterials, 2020, doi:10.3390/nano10112188_

Round 1
Reviewer 1 Report
The authors present a theoretical study about structural, electronic and optical properties of bulk, mono-, bi- and trilayered MX (M=Ga, In; X=S, Se, Te), and corresponding GaX/InX heterostructures.
They performed DFT based calculations, using the FP-LAPW method, as implemented in the program package WIEN2k. Several functionals for the exchange-correlation energy were applied. Such methodology is appropriate for the description of the stability and electronic property (e.g. band structure) The calculations seemed to be performed properly.
Though there exist already a number of theoretical studies about these systems. As correctly mentioned by the authors, the topological mechanisms of intralayer and interlayer bondings when vdW structures experience homo- and hetero-stackings remain unknown. The work is mainly devoted to this concern.
In contrast to many routine theoretical studies the authors investigate in detail the topological properties of electron density. Based on this analysis the relationship between microscopic interactions and macroscopic electromagnetic behaviors are studied for the first time on the individual layered Ga- and In chalcogenides and corresponding heterostructures.
Altogether, the study gives a clear and comprehensive picture of structure-property relationships of 2D Ga/In chalcogenides, as the authors have stated in the title of the paper.
The paper is very well written and illustrated. The relevant literature is considered sufficiently.
Therefore, I can recommend its publication in Nanomaterials in its present form.
Author Response
We are thankful to the reviewer for their careful analysis of our paper and for their recommendation.
Reviewer 2 Report
Review attached as a PDF file.

Author Response
Authors’ response (AR) to the reviewer:
- In the abstract, on line 14, vdW → van der Waals (vdW)
AR : The change in Abstract, line 15, has been made according to the reviewer’s comment, please see in the revised manuscript.
- In the Introduction, Our researches are of great meaning → Our research is of great meaning
AR : The change in the Introduction, line 79, has been made accordingly in the revised manuscript.
- In the Computational details, as well as the theoretical and experimental data. → as well as other theoretical and experimental data
AR : The change in the Computational details, line 90, has been made in the revised manuscript.
- I have a few questions about the following sentence: “The well-matched lattice constants have validated our parameter settings which will be used for the following layered and heterostructures calculations despite the undermined bandgaps.” When it says, “well-matched lattice”, what matches with what? When it says, “validated our parameter settings”, which parameters settings are being referred to? Is it the parameter setting of density functional or other details of the calculation? The phrase “undermined bandgaps” seems to imply that the band gap calculated from DFT is underestimated compared to the reference value but actually, the band gap obtained from mBJ (shown in Table 1) is not always underestimated compared to the reference. Hence it is not appropriate to say “undermined bandgaps”
AR : The “well-matched lattice” refers to the in-plane lattice constants matching between the calculated ones and other theoretical and experimental ones. The “parameters setting” we are referring to are the technical parameters, such as core-valence electrons configurations, muffin-tin radii, k-points, as well as energy, charge density, force convergence criteria, etc. Regarding the bandgaps, we agree with the reviewer, there are only underestimated with the WC-GGA and optB88-vdW functionals.
We have taken into account the reviewer’s comment and changed the phrase as follows:
“The good agreement between our calculated lattice parameters and literature ones validates our choices for the technical parameters settings, which will be used for the subsequent calculations on layered structures and heterostructures, despite the underestimated bandgaps obtained with the WC-GGA and optB88-vdW functionals.”
- In Figure 2a, it is very hard to see the numbers 1, 2, 3, etc. In Figure 3, the x-axis is very hard to see. Also, the fonts of figures in the supporting information is small. The fonts are in general small in all the figures.
AR : The fonts sizes in Figure 2a, Figure 3, Figure 5 and Figure 10 have been enlarged. Also, the fonts sizes in Supplement files have been adjusted.
Reviewer 3 Report
Interesting two-dimensional materials have been studied in this work. I have been shown that the band gap can tuned by varying number of layers in structure. It has suggested that separation of electrons and holes exists in studied heterostructures. Good work.
My several comments are listed below:
Page 1, Line 14:
Abbreviation “vdW” in this sentence should be replaced with: van der Waals (vdW).
Page 2, Figure 1.
Where did you get Figure 1? I can’t see any references in text.
Page 3, Line 85-86.
For bulk calculations, the first Brillouin zones were sampled with 1500 k-points using Monkhorst- Pack grids[39].
It is a really big value – 1500 k-points. Is this a typo? What k-mesh was used?
Page 4. Line 129-131.
For all cases, the direct (Γ-Γ’) and indirect (Σ-Γ) transition energy are close, which is consistent with the 130 results in Ref. [24]. 131
What does the “Γ’” means?
Author Response
- Page 1, Line 14: Abbreviation “vdW” in this sentence should be replaced with: van der Waals (vdW).
Authors' Response (AR) : The change has been made according to the reviewer’s comment.
- Page 2, Figure 1. Where did you get Figure 1? I can’t see any references in text.
AR : The results in Figure 1 are from our calculations. We understand that in general, the calculation results pertaining to the present work should not put in the introduction. As such, we have removed Figure 1 and related explanations from our revised manuscript.
- Page 3, Line 85-86. For bulk calculations, the first Brillouin zones were sampled with 1500 k-points using Monkhorst- Pack grids[39]. It is a really big value – 1500 k-points. Is this a typo? What k-mesh was used?
AR : In our calculations, the number of k-points has been optimized by varying it from 300 to 2400. As a result, the 1500 k-points is recognized as the best to obtain converged energy. The grid we used that corresponds to 1500 k-points is 20×20×3 k-mesh. We have added this information to the revised manuscript.
- Page 4. Line 129-131. For all cases, the direct (Γ-Γ’) and indirect (Σ-Γ) transition energy are close, which is consistent with the 130 results in Ref. [24]. 131 What does the “ Γ’ ” means?
AR : The symbol “ Γ’ ” is used to differentiate the position of conduction band maximum from the Γ of valence band minimum, when it comes to direct transition path. We have added this explanation information to the revised manuscript.